# Challenges and Opportunities for Drug Repositioning in Fibrodysplasia Ossificans Progressiva

**DOI:** 10.3390/biomedicines9020213

**Published:** 2021-02-19

**Authors:** Francesc Ventura, Eleanor Williams, Makoto Ikeya, Alex N. Bullock, Peter ten Dijke, Marie-José Goumans, Gonzalo Sanchez-Duffhues

**Affiliations:** 1Department de Ciències Fisiològiques, Universitat de Barcelona, IDIBELL, L’Hospitalet de Llobregat, 08907 Barcelona, Spain; fventura@ub.edu; 2Centre for Medicines Discovery, University of Oxford, Old Road Campus Research Building, Roosevelt Drive, Oxford OX3 7DQ, UK; eleanor.williams@cmd.ox.ac.uk (E.W.); alex.bullock@cmd.ox.ac.uk (A.N.B.); 3Department of Clinical Application, Center for iPS Cell Research and Application, Kyoto University, 53 Kawahara-cho, Shogoin, Sakyo-ku, Kyoto 606-8507, Japan; mikeya@cira.kyoto-u.ac.jp; 4Oncode Institute and Department of Cell and Chemical Biology, Leiden University Medical Center, Einthovenweg 20, 2333 ZC Leiden, The Netherlands; P.ten_Dijke@lumc.nl; 5Department of Cell and Chemical Biology, Cardiovascular Cell Biology, Leiden University Medical Center, Einthovenweg 20, 2333 ZC Leiden, The Netherlands; M.J.T.H.Goumans@lumc.nl

**Keywords:** activin, ALK2, BMP, bone, FOP, repurposed drug, signal transduction, TGF-β

## Abstract

Fibrodysplasia ossificans progressiva (FOP) is an ultrarare congenital disease that progresses through intermittent episodes of bone formation at ectopic sites. FOP patients carry heterozygous gene point mutations in activin A receptor type I *ACVR1*, encoding the bone morphogenetic protein (BMP) type I serine/threonine kinase receptor ALK2, termed activin receptor-like kinase (ALK)2. The mutant ALK2 displays neofunctional responses to activin, a closely related BMP cytokine that normally inhibits regular bone formation. Moreover, the mutant ALK2 becomes hypersensitive to BMPs. Both these activities contribute to enhanced ALK2 signalling and endochondral bone formation in connective tissue. Being a receptor with an extracellular ligand-binding domain and intrinsic intracellular kinase activity, the mutant ALK2 is a druggable target. Although there is no approved cure for FOP yet, a number of clinical trials have been recently initiated, aiming to identify a safe and effective treatment for FOP. Among other targeted approaches, several repurposed drugs have shown promising results. In this review, we describe the molecular mechanisms underlying ALK2 mutation-induced aberrant signalling and ectopic bone formation. In addition, we recapitulate existing in vitro models to screen for novel compounds with a potential application in FOP. We summarize existing therapeutic alternatives and focus on repositioned drugs in FOP, at preclinical and clinical stages.

## 1. Introduction 

Fibrodysplasia ossificans progressiva (FOP) is an ultrarare genetic musculoskeletal disease (OMIM: #135100). The prevalence of this condition is approximately 1 in 2 million worldwide with no indication of gender, ethnic, or geographical predisposition [1]. FOP progresses by episodic formation of ectopic bone (heterotopic ossification, HO), that arises from endochondral bone formation at extraskeletal sites including muscles, tendons, ligaments, and fascia. No HO is present in the tongue, diaphragm, and extraocular muscles, and both cardiac muscle and smooth muscle are also spared in FOP.

HO in FOP can be triggered by muscle injury following, e.g., biopsies, surgical interventions, or an accidental trauma, although an episode of HO typically develops following a flare-up [2]. Common features of a flare-up, which can last 6–8 weeks, are painful soft-tissue swelling, warmth, redness, stiffness, and decreased movement. A flare-up in FOP can arise spontaneously or be induced by minor soft-tissue injury, muscular stretching, falls, fatigue, intramuscular immunization, or an influenza-like illness. FOP flare-ups result in the transformation of soft tissue into heterotopic endochondral bone. Flare-ups usually follow specific anatomic patterns: they start in the neck, jaw, and shoulders, followed by the back, and advance to the trunk and limbs. The first episode of HO generally occurs during the first decade of life. FOP severely affects normal daily living due to the progression of HO throughout the body. HO in the trunk and extremities impairs motility and balance, increasing the chance of a fall causing trauma that exacerbates HO in those regions, rendering movement impossible in time. This leads to a vicious cycle of impairment and increasing disability. In addition to the progressive immobility, other life-threatening complications include severe weight loss following ankylosis of the jaw, affecting opening of the mouth and eating, as well as pneumonia and right-sided heart failure resulting from deformation in the thorax. This also makes breathing complicated and leads to the development of thoracic insufficiency syndrome [3]. Malformation of the great toe is also usually observed as a highly penetrant, congenital characteristic. Although HO is the major phenotype of FOP, atypical alterations in the central nervous system have also been reported, such as mild cognitive impairment, cerebellar abnormalities and hypoplasia of the brainstem [4,5]. Interestingly, brainstem lesions were observed early after birth, suggesting an impaired regulation of brainstem progenitor cell differentiation during development or early after birth [4,6]. Finally, although heart disease is typically overlooked in FOP, cardiac abnormalities in new-borns, such as ventricular septal hypertrophy, and cardiac conduction abnormalities have been reported [7,8], which might be related with vessel abnormalities [9,10,11] and endothelial dysfunction [12], also described in FOP.

In 2006, a heterozygous gain of function mutation (c.617G>A; R206H) in the glycine-serine (GS) domain of the activin A receptor type I (*ACVR1*), also known as activin receptor-like kinase 2 (ALK2), was identified as the genetic cause of FOP [13]. Approximately 97% of all FOP patients carry this single point mutation in *ACVR1*, but all 13 variant forms of *ACVR1* described to date seem to underlie the similar molecular cause of FOP (see later) [14]. Early diagnosis of FOP is crucial to limit disabilities and prevent iatrogenic harm, as to date there is no treatment proven to be effective for FOP to prevent flare-ups or HO. In the past, FOP was often diagnosed after the development of a flare-up, or even after the first evidence of HO. Currently FOP is diagnosed in young children prior to the onset of HO by identification of the malformation of the great toe, as previously described [15]. Surgical procedures are not suitable for the treatment of FOP, because the resulting trauma causes a far more serious HO [1,15]. Current treatment strategies for FOP aim to reduce inflammation and/or prevent ossification. Symptomatic use of corticosteroids such as prednisone [16] and nonsteroid anti-inflammatory drugs (NSAIDs), such as selective cyclooxygenase-2 (COX-2) inhibitors, are prescribed to relieve pain and prevent or delay HO in FOP, despite of the appearance of considerable side effects [17,18]. 

Currently, several molecules aiming to normalize the aberrant function of the mutant *ACVR1* are in ongoing clinical trials for the treatment of FOP (https://www.ifopa.org/ongoing_clinical_trials_in_fop, accessed on 11 January 2021). In this manuscript, we will discuss the aberrant signal transduction pathways altered by mutations in *ACVR1*/ALK2 and briefly review the existing in vitro platforms to screen for novel drugs and genes with therapeutic potential in FOP. We will introduce the existing treatments under clinical investigation and focus on drug repositioning as an alternative promising approach with potential benefits in the field of rare diseases.

## 2. Aberrant TGF-β Signalling Underlies FOP

As previously mentioned, in past years genetic studies have revealed the presence of single point mutations in the gene encoding the bone morphogenetic protein (BMP) type I kinase receptor ALK2. Using more advanced in vitro systems expressing the mutant ALK2 (see below), we now know that FOP involves disturbances in BMP and transforming growth factor (TGF)-β-like signalling. BMPs, TGF-βs, and other related molecules like activins are circulating and locally acting growth factors belonging to the transforming growth factor (TGF)-β family. Therefore, in order to understand aberrant signal transduction in FOP, we will briefly introduce the different signalling cascades of TGF-β family members.

TGF-β ligands are structurally related secreted dimeric cytokines that act in an autocrine, paracrine, and endocrine manner on a large and diverse spectrum of cell types [19]. They exert pleiotropic cellular functions of which the effects are highly dependent on cellular context [20,21]. Not only do they play pivotal roles in multicellular organisms during embryogenesis and in maintaining tissue homeostasis, but their malfunction has been causally associated with developmental disorders, cancer, cardiovascular, and muscle-skeletal diseases [22,23,24,25,26]. TGF-β family members signal via structurally related complexes of single transmembrane spanning TGF-β family type I and type II receptors that are endowed with an intrinsic serine/threonine kinase domain [27]. Five human type II receptor and seven human type I receptors have been identified. The type I receptors are also termed ALKs. Each ligand within the TGF-β family has its selective partner receptors. For example, whereas activin binds to activin type II receptors (ActRII and ActRIIB) and ALK4 and ALK2, BMPs interact with ActRIIs and BMP type II receptor (BMPR2) and ALK1, ALK2, ALK3, and ALK6 [27,28]. Different TGF-β family members can exert similar functions but also distinct functions, and even antagonize each other’s functions. As described, activins and BMPs share common receptors and several recent studies have demonstrated competition of these ligands for these receptors as an important mechanism for crosstalk [29,30,31]. 

To initiate intracellular signalling, the TGF-β family soluble ligand induces heteromeric type I/type II membrane receptor complex formation, upon which the constitutively active type II kinase trans-phosphorylates serine and threonine residues in the juxta-membrane domain of the type I receptor, also called glycine and serine residue rich (GS) domain) [27,28]. This activates the type I receptor, which subsequently induces the phosphorylation of intracellular substrates that include the SMAD transcriptional proteins [32]. Whereas activation of TGF-β type I receptor (ALK5) and ActRIB (ALK4) mediates the phosphorylation of receptor-regulated SMAD2 and SMAD3, BMP type I receptors (ALK1, ALK2, ALK3, and ALK6) induce SMAD1, SMAD5, and SMAD8 phosphorylation. The type I receptor-induced SMAD phosphorylation occurs on the two most carboxy terminal serine residues. Activated receptor-regulated SMADs partner with the common mediator SMAD4. These heteromeric complexes translocate to the nucleus where they participate in transcriptional responses by cooperating with co-activators and co-repressors [33]. Receptor-regulated SMADs (except for the predominant spliced form of SMAD2) and SMAD4 bind in a sequence-dependent manner to DNA. The affinity is rather weak, and there is a need for cooperation with other DNA-binding transcription factors [33]. 

Besides the canonical SMAD pathway, TGF-β family receptors can also initiate noncanonical signalling that is SMAD-independent [34,35]. Examples of these pathways include extracellular signal-regulated kinase (ERK), p38, and c-Jun amino terminal kinase (JNK) mitogen-activated protein (MAP) kinases, RHO family GTPases, phosphatidylinositol-3 kinase (PI3K), and nuclear factor-kappa B (NF-κB) signalling. These non-SMAD signalling pathways are not specific for TGF-β family members and occur in a cell-type-dependent manner. Interestingly, some of these pathways, such as the E3 ubiquitin ligase tumour necrosis factor receptor-associated factor (TRAF)-4-mediated activation of TGF-β activated kinase-1 (TAK1), which can trigger JNK/p38, PI3K, and NF-κB signalling, occur in an ALK5-dependent but kinase-independent manner [36,37,38,39]. The SMAD and non-SMAD pathways crosstalk and fine-tune each other’s responses [34,40]. For example, MAP kinases (MAPKs) can phosphorylate SMADs and either inhibit [41] or potentiate SMAD function [42,43]. MAPKs induce activation of activator protein-1 (AP1) transcription factors, which can partner with SMADs and thereby enforce or repress certain transcriptional responses [44,45,46,47,48]. In addition, activated SMADs can transcriptionally induce the expression of ligands of growth factors that act through tyrosine kinase receptors. Thus, they can, in an SMAD-dependent manner, elicit non-SMAD signalling responses [34]. Of note, these findings indicate that pharmacological modulation of the TGF-β family type I receptor kinases or enzymes involved in non-SMAD signal pathway may perturb but not block all signalling responses. 

The pathological mechanisms underlying FOP involve an imbalance between BMP and TGF-β signalling, including canonical and non canonical signalling. Therapeutic agents for FOP should aim to normalize the aberrant ALK2 signalling pathway, either directly or by affecting pathways that crosstalk with it. Inflammation plays an important contribution exacerbating the mutant ALK2-induced HO response. In the next section, we will discuss how ALK2 signal transduction becomes dis- and mis-regulated in FOP.

### ALK2 Signalling in FOP

Since the discovery of *ACVR1* as a causative gene for FOP, different in vitro models resembling aberrant ALK2 signalling have been used to dissect the molecular and cellular mechanisms underlying FOP. In this section, we will discuss the current knowledge regarding the malfunction of ALK2 in FOP.

Although different mutations in the *ACVR1* gene (encoding ALK2) have been identified, the heterozygous single point mutation in the GS region leading to ALK2^R206H^ is present in more than 97% of FOP patients [49,50]. ALK2^R206H^ displays mild ligand-independent BMP signalling, as well as increased responsiveness to BMP stimulation. In addition to the c.617G>A *ACVR1* mutation (encoding ALK2^R206H^), all mutations detected in FOP and diffuse-intrinsic pontine glioma (DIPG) also show enhanced SMAD1/5/8 responses to BMPs [51,52,53,54,55,56,57]. Interestingly, the mutant ALK2 does not require its ligand-binding domain to over activate BMP signalling [56]. Therefore, it has been suggested that the overactivation of the mutant ALK2 could arise from a reduced ALK2 interaction with the intracellular negative regulator FKBP prolyl isomerase 1A (FKBP12). FKBP12 binds to the GS domain of ALK2 and suppresses its leaky activation, by spontaneous non ligand mediated type I/type II complex formation [58,59,60,61]. In the presence of a ligand, phosphorylation of the GS domain by type II receptors releases FKBP12, allowing type I receptor activation [62,63]. In fact, crystallographic studies found that *ACVR1* mutations in the GS domain of ALK2 directly impair binding to FKBP12 and resulted in its basal overactivation [52].

Recent studies added compelling evidence that the ALK2^R206H^ mutation also confers an unexpected signalling neofunction. ALK2 was originally identified as a cell-surface protein interacting with TGF-β and activin A [64,65]. Subsequently, ALK2 was found to bind to and signal for BMPs (such as BMP-5/6/7), thereby triggering SMAD1/5/8 activation [66,67,68,69]. Several consecutive studies confirmed that ALK2 forms a non signalling complex upon binding to activin A and type II receptors [30,70], and that activin and BMP7 compete for type II receptor binding [29]. In FOP, ALK2^R206H^ gained the ability to induce SMAD1/5/8 activation upon activin A stimulation [54,71] and subsequently to trigger endochondral ossification and enhance chondrogenesis of induced mesenchymal stromal cells derived from FOP induced pluripotent stem cells (iPSCs) in vitro [54,71]. Additionally, activin A also induced SMAD1/5/8 activation through ALK2 in other atypical FOP-causing *ACVR1* mutations [51,54]. Consistent with this notion, spontaneous and trauma-induced HO were inhibited or partly reduced in most mice treated with activin A-neutralizing antibodies [72].

The exact mechanism by which mutations in the intracellular domain of ALK2 cause a switch for activin A-induced ALK2 receptor complexes from a non signalling to a BMP-signalling configuration remains elusive [73,74]. While heterozygous mutations in *ACVR1* cause FOP, complete loss of the wild-type *Acvr1* allele in *Acvr1*^R206H/+^ mice results in a substantial increase in HO volume [72]. ALK2^R206H^ requires its type II receptor partners, BMPR2 and ActRIIa to signal after engaging either BMP-6/7 or activin A, but it does not seem to require their type I receptor partners ALK3 or ALK6 [54,56]. Pointing also to the importance of the relative balance between receptors, some studies reported that activin A signalling through wild-type ALK2 occurred in a cell-type-dependent manner [30,75]. The effects of BMPR2 on SMAD1/5 signalling can be paradoxical: the loss of BMPR2 potentiated, and BMPR2 overexpression reduced, the SMAD1/5/8 signalling via ALK2-ActRIIa/b induced by activin A [30]. Altogether, this evidence highlights the importance of competition between wild-type and mutant ALK2 receptors. This competition for complex formation preference might determine the signalling outcome, which is further altered in the presence of ALK2^R206H^ [30,76].

Non canonical ALK2 signalling also plays a role in FOP. For instance, lymphocytes derived from FOP patients have dysregulated ALK2-p38 MAPK signalling that can be blocked with p38 inhibitors [77]. The phosphatidyl-inositol 3-kinase (PI3K)/AKT/mechanistic target of rapamycin (mTOR) pathway has also been linked to trauma-induced HO and FOP. Different inhibitors of mTOR complexes, including rapamycin, reduced ossification in FOP mice [78,79], similarly inhibitors of PI3Kα prevented SMAD activation and HO in animal models of FOP [80]. Moreover, mutant *Acvr1*^R206H/+^ mice also showed increased activation of RHOA, altered cell morphology, and misinterpretation of the tissue microenvironment [75].

In summary, during the last 15 years the molecular mechanisms underlying ectopic bone formation in FOP have started to become unveiled, enabling further research into drug development. Gathering this information has been partially possible through the establishment of in vitro models, in which BMP and, more specifically, ALK2 signalling is disturbed to resemble FOP. In the next section, we briefly describe the different and recently developed in vitro models available to identify potential new targetable mechanisms and compounds with therapeutic potential in FOP.

## 3. In Vitro Research Platforms Resembling FOP

During the past years, new mechanistic insights have led to the identification of potential therapeutic targets in FOP. Patient-derived induced pluripotent stem cells (iPSCs) and transgenic mice expressing the mutant *ACVR1* gene in a tissue and time restricted manner are now extensively used in the field. In addition to such recent developments, various in vitro models with increasing degrees of complexity have been established and utilized historically (Figure 1). These models were developed to resemble particular hallmarks of FOP, such as increased ALK2 activation, exposure to inflammation or bone-cell differentiation. Furthermore, one might make use of in vitro/ex vivo models to investigate tissue-specific responses to TGF-β family ligands and drugs (for example, cardiotoxicity tests in iPSC-derived cardiomyocytes or endothelial cells [81]). 

For drug-development strategies, particularly drug repositioning, such models are useful in early preclinical phases to screen for approved candidate drugs with therapeutic potential. Depending on the biological process to be targeted (e.g., SMAD1/5/8 activation, bone matrix deposition), a simple in vitro assay system may be preferred. In order to facilitate the selection of a model, in this section, we will summarize and briefly discuss the pros and cons of existing in vitro models for FOP.

Even before the discovery of genetic mutations in *ACVR1* in patients with FOP, aberrant BMP signalling activity was linked with FOP disease progression [83,84]. Therefore, expression levels of BMPs (e.g., BMP4) were monitored as a surrogate for bone activity [85], mainly using patient-derived lymphoblastoid lines isolated from peripheral blood. Researchers introduced the use of recombinant BMP ligands to stimulate cell lines lacking the mutant receptor (e.g., C_2_C_12_ [86], PASMCs [87], QM7 [55]). One should note, however, that depending on the relative expression level of every BMP receptors (and co-receptors) and the concentration of the ligand, BMP ligands may signal via BMP receptors other than ALK2 (reviewed in [88]). This becomes relevant when searching for ALK2-specific antagonists or downstream pathways, for example. 

Following the association of *ACVR1* single point mutations with FOP, in vitro models were developed in which ALK2^R206H^ and other less-common mutant receptors were over-expressed both transiently and stably. Experiments performed under these conditions rapidly highlighted that mutations in the intracellular GS or kinase domain of ALK2 rendered the receptor ligand-independent and constitutively active [59,60,86,89,90]. In addition, these studies allowed for comparison between different ALK2 mutations in FOP, as well as with the artificial constitutive active form of ALK2^Q207D^ (thus far, not reported in humans). In order to fairly compare the effects between different versions of ALK2 using these overexpression strategies, it becomes key to confirm (and normalize) the expression levels of the ectopic genes. Moreover, non physiological levels of expression may lead to experimental artefacts (for example, non spontaneous association and aberrant overactivation of the kinase receptors), which should be taken into account. In part to overcome these disadvantages, new cell lines with endogenous expression of mutant ALK2 were established. This was possible, for example, differentiating patient-derived (iPSC) or inserting the FOP gene mutation in non FOP lines, using regularly interspaced short palindromic repeats (CRISPR)/Cas9 gene-editing approaches [54,79,91].

The development of mouse models of FOP facilitated the establishment of immortalized murine embryonic fibroblasts (MEFs [92]) and, more recently, fibroadipogenic progenitor cells (FAPs) which have been demonstrated to drive HO autonomously [72], resembling the ability to differentiate into the osteo/chondrogenic lineage. These cells, therefore, may represent a suitable tool for in vitro functional assays. Moreover, Mx1^+^ (MX dynamin-like GTPase 1) and Scx^+^ (Scleraxis) cells were freshly isolated by flow cytometry from muscles of transgenic *FOP* mice [93]. These two cell populations displayed enhanced alkaline phosphatase (ALP) activity in vitro, when stimulated with recombinant activin A. 

A number of patient-derived primary cell systems have been utilized during the last decades for FOP research. As such, mutations in *ACVR1* were identified in DNA sequencing in peripheral blood lymphoblastic lines from FOP donors [13]. Furthermore, peripheral blood monocytes have been used to profile immune cell populations in FOP [84,85,94,95]. Skin fibroblasts cultured from 3-mm-thick skin biopsies were reported to exhibit enhanced SMAD1/5 activation in the presence of serum and increased expression of osteogenic markers when cultured under osteogenic conditions [96]. Periodontal ligament cells from extracted teeth were used to test the effect of the small molecule TGF-β type I and type II kinase inhibitor GW788388 [97] comparing control and FOP donors. Periodontal ligament cells could give rise to osteoblast- and osteoclast-like cells in vitro [98] and respond effectively to ectopic activin stimulation by increasing BMP receptor downstream targets [99]. A population of peripheral blood circulating endothelial cells, named endothelial colony forming cells (ECFCs), has been recently characterized in FOP patients. ECFCs exhibit aberrant phospho-SMAD1/5/8 activation in response to activin. Moreover, when cultured under osteogenic conditions, these cells represent a functional model to analyse osteogenic differentiation in FOP [12].

A remarkable advance in the development of in vitro research models for FOP has been achieved through the generation of induced pluripotent stem-cell (iPSC) lines. iPSCs are somatic cells reprogrammed by the overexpression of four transcription factors (octamer-binding protein 4 (OCT4), SRY-Box transcription factor 2 (SOX2), Kruppel-like factor 4 (KLF4) and MYC proto-oncogene (c-MYC); the so-called Yamanaka factors) [100,101]. Because of their pluripotency and unlimited ability to proliferate, iPSCs have multiple medical applications including regenerative medicine, disease modelling, and drug discovery using either iPSCs from healthy donors or patients [102,103]. To date, different somatic sources (e.g., skin fibroblasts, SHED cells, renal cells, and periodontal ligament fibroblasts) and reprogramming strategies have been used to obtain stable iPSC colonies with pluripotent differentiation potential and unlimited doubling capacity [104,105,106,107,108]. Due to the variability between differentiation batches, the use of isogenic rescued lines where the *ACVR1* mutation has been corrected by means of CRISPR/Cas9 is highly recommended. This strategy has been recently followed to gain deeper understanding into the molecular mechanisms driving FOP [54,109,110] but also to identify novel molecules with specific activity against the mutant receptor [79]. Moreover, differentiation protocols can be optimized in order to faithfully resemble specific cell-type phenotypes in FOP, including fibroadipogenic progenitor (FAP) cells [109].

Irrespective of the experimental model chosen, commonly used functional readouts in these assays include phosphorylation of SMAD1/5, target gene expression (e.g., *Inhibitor of DNA-binding 1 (Id1), Id2, Id3)* and/or transcriptional reporter assays containing BMP SMAD responsive elements. TGF-β family members other than BMPs regulate bone growth and homeostasis (e.g., TGF-βs, activins), and different branches of the TGF-β signalling pathway tend to balance one another through positive and negative feedback mechanisms. Furthermore, TGF-β signal transduction very often crosstalks to other signalling pathways relevant in bone tissues (e.g., Wnt, Yes1 associated transcriptional regulator (Yap)/Tafazzin (Taz), fibroblast growth factor (FGF)). In this sense, one should note that inflammatory signalling plays a major role inducing episodes of genetic and trauma-induced HO [111]. In vitro models allow one to mimic a pro-inflammatory environment by, for example, adding exogenously inflammatory cytokines (e.g., tumour necrosis factor [TNF]-α, interleukin [IL]-1β)) to the cells in culture [12]. In order to elucidate the complex crosstalk and feedback in FOP cells, it is therefore advisable to obtain comprehensive transcriptomics and (phospho)proteomics expression signatures in combination with functional studies when investigating the effect of a particular gene or pharmacological treatment in FOP.

Enhanced BMP signalling also induces in vitro osteogenic differentiation of different cell types including endothelial cells, myogenic cells, interstitial mesenchymal cells, and likely FAPs and tendon-derived progenitor cells [60,72,93,112,113]. Therefore, functional osteogenic (and chondrogenic) differentiation assays can be performed, preferably in cells with endogenous expression of ALK2^R206H^, to test the effect of potential drugs or genes in FOP.

In vitro models provide a relatively easy and simple methodology to identify and select drug candidates with enhanced specificity and low cell toxicity. In the case of repositioned drugs, pharmacokinetic characterization has been performed already and normally compound libraries contain molecules with good drug-like properties (e.g., solubility, metabolic clearance). Subsequently, selected candidates are validated using in vivo animal models of the disease, where, for example, the mobility of the animals is studied and the effect of the drug on HO volume is monitored by histo-morphometric techniques, including X-ray and/or micro-CT analysis. In the case of FOP, different models have been published including BMP-induced ectopic ossification [114] and ALK2-mediated HO, with either the transgenic mice bearing the constitutively active mutant receptor ALK2^Q207D^ [87,89] or the humanized ALK2^R206H^ [115] expressed in an inducible and tissue-specific manner [72,93,112].

In part due to the interplay and overlap between the ALK2 signalling pathway and other signal transduction cascades, a number of novel pharmacological options to treat ectopic bone formation in FOP have emerged in recent years from the repurposing of existing medications.

## 4. Repurposed Drugs for FOP

As previously mentioned, it was shown that mutant receptor inactivation (by either targeting its kinase activity or ligand interaction) successfully prevented osteogenic differentiation in vitro and HO in animal models of FOP. Therefore, a number of academic and corporate teams have pursued the development of specific kinase inhibitors for ALK2^R206H^, some of which have entered into clinical trials. However, due to the high structural similarity in the kinase ATP pocket, especially among TGF-β, activin, and BMP receptor kinases, it is challenging to identify specific molecules suitable for further clinical investigation (reviewed in [88]). Alternatively, receptor inactivation may be achieved by preventing ligand interaction and/or promoting receptor internalization, or by inhibiting oligomerization with type 2 receptors. Of note, a humanized anti-activin A neutralizing antibody (REGN2477) has recently progressed into Phase II clinical trials (NCT03188666; NCT04577820), and an anti-ALK2 antibody toward mouse ALK2 extracellular domain-Fc protein has recently been patented (Daiichi Sankyo/Saitama, US patent application 10428148). 

While we still wait for the outcome of these studies, a new class of drug candidates for FOP has entered the race to become a valid therapy for FOP. Repurposed drugs arise from the application of existing validated drugs to a different disease indication. In particular for rare diseases such as FOP this appears an attractive option. 

### 4.1. Drug Repurposing versus De Novo Drug Development

Compared to the development of new drugs, drug repurposing has a number of advantages (summarized in Table 1). As less than 15% of new therapeutic compounds reach approval [116], drug repurposing substantially reduces the risk of failure, whilst also reducing costs in drug development and shortening the time to clinical implementation [117,118]. This is an attractive benefit for rare diseases such as FOP, with a very limited market. There are multiple examples of drugs that have been repositioned for new indications, including the hypertension drugs sildenafil [119] and minoxidil [120], which have new indications in erectile dysfunction and hair growth, respectively. Thalidomide was repositioned from the treatment for morning sickness to multiple myeloma [121]. Rituximab used for treatment of various cancers was repurposed for Rheumatoid arthritis [122] and use of imatinib for therapy of BCR-ABL-positive cancers was expanded to c-KIT gastrointestinal cancers [123]. More recently, remdesivir (a broad spectrum antiviral) was repurposed for treatment of COVID-19 [124]. There are several drug repurposing chemical libraries that have been described in literature that include (pre)clinical compounds. An example of such a drug-repurposing library is one from the Broad Institute that holds 5685 small molecule therapeutics [125]. These are compounds with safety data that have excellent pharmaco-dynamic and -kinetic profiles that can be readily tested in animal (disease) models and rapidly developed for clinical use in a new indication. 

When high-throughput screens are conducted for a certain target or biological phenotype, the target may be different from the one previously described (so-called off target). The identification of off-target protein binding can be performed for example by mass-spectroscopy-based thermal proteomic profiling [126]. If a drug is available for the newly identified target, then this can be compared side by side with the hit compound from the repurposing library. In addition, the compound can be chemically modified, and inhibition and selectivity optimized for the newly identified target. As patent protection can likely be obtained, the latter approach may also increase the enthusiasm from industry to invest and develop the modified compound for the desired clinical indication. 

During recent years, a number of repositioned drugs have been demonstrated to make effective treatments for FOP at preclinical stages, and recently some of them have entered in human clinical trials. We will briefly discuss some of these examples, including compounds in a preclinical stage and dedicate a special focus on saracatinib and rapamycin (Figure 2).

### 4.2. Preclinical Candidates

Years of intense basic and clinical research in FOP have led to the identification of major cellular and molecular processes essential for the progression of HO. Therefore, available drugs able to target these pathways became appropriate candidates for further preclinical research. In addition to the aforementioned drugs designed to target ALK2 or its ligands, several repurposed drugs emerged as new potential therapies for FOP and are currently in preclinical development. FOP progression was rapidly associated to inflammatory events. Therefore, one of the first approaches envisaged was to reduce the immune response and inflammation. In fact, corticosteroids are currently a main therapeutic strategy to manage flare-ups [127]. In addition to anti-inflammatory steroids, from distinct NSAID molecules tested, celecoxib has been shown to reduce HO after trauma in a rat tenotomy model, and the COX-2 inhibitors celecoxib and etoricoxib partially prevented HO after total hip replacement [128,129,130,131]. However, there is still a lack of evidence that chronic treatment with NSAIDs prevents flare-ups in FOP patients.

It has been shown that hypoxia promotes FOP, partially by prolonging BMP signal duration, and becomes indispensable for early chondrocyte differentiation [132,133]. In accordance, several hypoxia-inducible factor (HIF)1α inhibitors, such as PX-478, apigenin, or imatinib strongly prevented HO in mouse models of FOP by inhibiting the formation of mesenchymal condensations [134,135]. From these molecules, imatinib, developed for treatment of chronic myeloid leukaemia, has a long-standing record of minor side effects in adults and children [136]. Imatinib was originally designed as a tyrosine kinase inhibitor of the BCR-ABL fusion protein. Later, it has been demonstrated to also affect multiple pathways that are important in the inflammatory and early hypoxic stages of FOP. For instance, it inhibits the HIF1α, platelet-derived growth-factor receptor (PDGFR), KIT Proto-oncogene, receptor tyrosine kinase (c-KIT), and multiple MAPKs, thereby playing immunosuppressive effects in lymphocytes, macrophages, and mast cells [137]. Based on this evidence, imatinib was prescribed in an off-label basis at a non trial setting in seven children with continuous flare-ups, who were refractory to standard-of-care treatments [138]. All six patients that took imatinib regularly reported decreased intensity in their flare-ups. Since the study relied on retrospective anecdotal reports, there is no conclusive evidence of the beneficial effects of imatinib. Future clinical trials should provide such evidence about the usefulness of imatinib in preventing HO.

The anti-anginal agent perhexiline was found in a screening of FDA-approved drugs for their ability to block *Id1* induction upon ALK2^R206H^ activation [139]. However, although perhexiline was able to induce a slight reduction in BMP-induced HO in a murine model, an open-label study with five patients could not demonstrate the efficacy of oral perhexiline administration in the prevention of HO in FOP [140].

Palovarotene is a highly specific retinoic-acid receptor (RAR)-γ agonist which has been under clinical investigation for several conditions (e.g., emphysema [141,142,143], hereditary multiple exostoses NCT03442985). Later studies highlighted a potential application of this molecule to inhibit HO in FOP, by directly targeting endochondral ossification and SMAD1/5 expression [144,145], which led to the initiation of clinical trials in FOP. Currently, the beneficial effect of palovarotene in FOP is under investigation by the authorities (see below).

Recently, it has been found that inhibitors of phosphatidyl-inositol 3-kinase alpha (PI3Kα) could become a useful therapy for patients suffering from FOP. PI3Kα inhibitors likely target different cell types required for HO lesion progression. PI3K/AKT was shown to be a potent target to down-regulate mast cell function and, in turn, reduce the severity of mast cell-dependent diseases [146]. Alpelisib (also known as BYL719 and marketed as Piqray^™^) prevented HO in vivo in a murine model of FOP and inhibited canonical and neomorphic ALK2^R206H^ responses in murine pluripotent cells and iPSCs from FOP patients [80]. Alpelisib was recently approved by the FDA for treatment of patients with hyperactive PI3K (PIK3CA)-altered solid tumours [147]. In addition, it has been shown to be clinically effective in patients with PIK3CA-related overgrowth syndrome (PROS) [148]. The recommended dose of alpelisib approved by the FDA for oncological purposes is 300 mg daily. Daily oral doses of 400 mg of alpelisib were well tolerated by patients in a phase III study in PIK3CA-altered solid tumours [149] whereas PROS patients, after eighteen months, are still being treated daily with 250 mg of alpelisib [148]. In these studies, the most frequent adverse effects were hyperglycaemia and rash, an expected effect of PI3Kα inhibition that could be managed by concomitant metformin or preventive anti rash treatments [149]. Alpelisib reduces chondrocyte and osteoblast commitment of mesenchymal stem cells and mice deficient for PI3Kα in osteoblasts develop osteopenia [150]. Furthermore, expression of activated AKT in transgenic mice promoted chondrocyte differentiation, whereas a dominant-negative form delayed this process [151]. Angiogenesis also requires PI3Kα activity to control endothelial cell migration [152], therefore alpelisib may prevent vascular recruitment within HO lesions. Mechanistically, PI3Kα inhibitors can negatively target ALK2 kinase activity [153] and increase SMAD1/5 degradation by affecting of GSK3 activity, reducing BMP transcriptional responses [154]. Moreover, because GSK3-regulated cellular levels of β-catenin are controlled through a coordination of PI3K and WNTs signalling, we can speculate that canonical WNT responses may be also blunted by alpelisib. In addition, mTOR, also downstream of PI3K/AKT, has been shown to be an important pathway for FOP and nongenetic HO (see below) [78,155]. Therefore, targeting PI3Kα/AKT has the potential to suppress HO by the inhibition of SMAD, mTOR, and WNT signalling pathways, which are essential for the early inflammatory and the late osteogenic phases. Although these data are encouraging, to set a solid base for a clinical treatment, further studies are required to optimize alpelisib treatment in both episodic and spontaneous HO in FOP.

### 4.3. Saracatinib

Saracatinib (AZD0530) is a small-molecule kinase inhibitor developed by AstraZeneca UK Limited. Originally identified as a selective dual inhibitor of the Src/Abl kinases for indications in oncology, saracatinib shows excellent pharmacokinetic properties with good oral bioavailability and a half-life of ~40 h [156]. As such, saracatinib has been investigated in over 30 phase I and II clinical trials involving the dosing of over 700 patients. In phase I trials, saracatinib showed a maximum tolerated repeat dose of 250 mg [157] and tolerability up to 125 mg for longer-term chronic dosing [158]. It was subsequently investigated in phase II trials for conditions including ovarian cancer [159,160], small-cell lung cancer [161], prostate cancer [162], colorectal cancer [163], and pancreatic cancer [164]. However, to date these trials have reported insufficient efficacy for saracatinib to be investigated further in phase III as a treatment option in oncology.

In a pioneering initiative, rather than abandoning this asset, AstraZeneca established collaborations with the Medical Research Council in the United Kingdom, Europe’s Innovative Medicines Initiative and the National Center for Advancing Translational Sciences of the National Institutes of Health in the United States to make this and other investigational drugs available to academics for grant-funded experimental medicine studies [165]. Through such partnerships, phase II studies were recently conducted using saracatinib as a Src inhibitor for the treatment of lymphangioleiomyomatosis (NCT02737202) [166] and as a Fyn kinase inhibitor for the treatment of Alzheimer’s disease (NCT02167256) [167]. Both studies used a lower daily dose of 100 to 125 mg saracatinib as a tolerated drug regime for treatment over a longer period of 9 to 12 months.

The potential for saracatinib to be used as a treatment for FOP was uncovered recently by two independent studies [79,168]. Williams et al. [168] performed an in vitro screen of 150 clinically tested kinase inhibitors to identify those compounds binding to the recombinant ALK2 kinase domain. Saracatinib was identified as the most potent ALK2 inhibitor in the screen (IC_50_ = 6.7 nM) and was found to be relatively selective for ALK2, ABL, and Src family kinases when screened against a larger panel of 252 human kinases [168]. Approved drugs based on the same quinazoline scaffold, such as gefitinib [169], showed little-to-no ALK2 binding suggesting that the pendant moieties in the derivative saracatinib were particularly compatible with ALK2. This was confirmed by the crystal structure of ALK2 bound to saracatinib (PDB ID: 6ZGC) which showed the expected binding of the quinazoline to the ATP-binding pocket of the ALK2 kinase domain as well as favourable hydrophobic interactions for its pendant chlorobenzodioxole moiety [168]. In cellular reporter assays, saracatinib showed selectivity for inhibition of BMP over TGF-β signalling consistent with its selectivity for ALK2 over ALK5. The neoactivity of mutant ALK2 signalling in response to activin A was also inhibited. Based on these in vitro results, saracatinib was tested as a prophylactic treatment in two Cre-inducible mouse models expressing ALK2^Q207D^ [87,170] or ALK2^R206H^, respectively [71]. In both cases, saracatinib prevented the development of HO and preserved the range of motion with no detected effect on neonatal growth [168].

The second study by Hino et al. [79] identified saracatinib from a library of 5000 compounds using a cell assay based on the induction of alkaline phosphatase activity following expression of ALK2^R206H^ in the chondrogenic cell line ATDC5. An orthogonal cell assay in FOP patient-derived iPSCs further showed that saracatinib could suppress chondrogenesis upon activin A stimulation. The compound was then tested in three different mouse models of HO, including cardiotoxin-induced injury in a transgenic mouse model expressing ALK2^R206H^, [78] a BMP7-induced model [78], and a humanised FOP model using FOP-iMSC’s and activin-A-expressing cells transplanted into mice [54]. In all models, saracatinib was found to suppress HO formation compared to controls [79].

An academic investigator led the phase II trial to determine the safety and efficacy of saracatinib in FOP patients was launched in 2020 funded by Europe’s Innovative Medicines Initiative (IMI2). The STOPFOP trial (NCT04307953) will test saracatinib in adult FOP patients through a 6-month double-blind randomised control study followed by an open label extension phase. Patients will be monitored via low-dose whole-body CT to evaluate the total change in heterotopic bone volume.

### 4.4. Rapamycin

Rapamycin is an approved drug, and its safety has been shown after nearly 20 years of use. Rapamycin was first isolated from the actinomycete *Streptomyces hygroscopicus* in the soil of Easter Island [171]. Although it was initially used as an antifungal drug, it was later found to have strong immunosuppressive and anti-proliferative effects and is now used to prevent transplant rejection and lymphangioleiomyomatosis. Inhibition of mTOR signalling has been identified as the main molecular mechanism underlying its preventative action. Indeed, the name mTOR directly refers to this effect: mechanistic target of rapamycin (originally, the “m” stood for “mammalian”).

Rapamycin has been recently identified as a potential drug for the treatment of FOP. An in vitro screening model was established by generating isogenic iPSCs from FOP fibroblasts, which were subsequently differentiated into mesenchymal cells (MSCs) [107,108]. This model was used to screen for chemical compounds to suppress the enhanced chondrogenesis of FOP and rescued iPSC-MSCs, by testing an original drug repurposing-focused library, which contained approximately 7000 small-molecule compounds. Rapamycin was identified as a candidate drug that suppresses the enhanced cartilage formation [78].

Rapamycin has been reported to suppress trauma-induced and constitutively active ALK2^Q207D^-induced ectopic bone formation [135]. The proposed mechanism suggests HIF1α inhibition, which leads to the down-regulation of SRY-Box transcription factor 9 (Sox9), the master transcription factor of cartilage. Activin A abnormally transduces BMP signalling through mutant ALK2 receptors and induces the expression of *ENPP2* to activate mTOR signalling and thus abnormal chondrogenesis [54,78]. In this FOP iPSC-based model, the expression level of HIF1α was up-regulated during chondrogenesis. However, it remained similar between FOP iPSC-derived cells and genetically rescued iPSC-derived cells, suggesting that HIF1α up-regulation is a common feature of chondrogenesis and that this is not a FOP-specific phenomenon.

Notably, mTOR signalling is multifunctional and controls many molecular and cellular events besides HIF1α translation [171]. For example, mTOR signalling activates T and B cells, which may be associated with the early phase of the FOP pathology. It induces cell proliferation through the promotion of mRNA splicing and lipid synthesis and also suppresses autophagy. Cell proliferation is a phenomenon observed in the early stages of ectopic ossification, and its suppression by rapamycin may reduce the FOP pathology. These anti-inflammatory effects of rapamycin have been reported previously in clinical trials, including in children [172,173]. Nevertheless, studies have revealed a number of potential side effects too, which may need to be carefully monitored in FOP patients. Following these promising results, an academic-initiated phase 2 clinical trial for a 6-month randomized placebo-controlled study and subsequent open-label extension study began in September 2017, including 20 FOP patients in Japan (UMIN000028429).

## 5. Conclusions and Perspectives

FOP is an ultrarare congenital disease that progressively debilitates affected individuals and puts them under high risk of life-threatening complications. In the last decades, enormous advances have been achieved, such as the identification of the causative gene and certain underlying molecular mechanisms, the proper characterization of the disease progression through natural history studies, the development of in vitro and in vivo models resembling specific features of FOP, and the identification of specific cell types involved in HO and potential molecules with therapeutic potential. However, despite these efforts, to date there is no validated cure or biomarker for this terrible disease.

A number of potential treatments are currently under investigation (www.ifopa.org). Palovarotene (Ipsen) is at this moment in a phase III clinical trial (MOVE: NCT03312634). After phase II study (NCT02190747) showed a decrease in flare-ups and HO, the MOVE trial was initiated. Unfortunately, the phase III study was temporarily interrupted in January 2020 due to lack of efficacy and potential side effects reported in a new juvenile mouse model of FOP [174,175,176]. REGN2477 is an antihuman activin-A-neutralizing antibody, also known as garetosmab (Regeneron), investigated for its efficacy and safety for the treatment of FOP in a phase II randomized, placebo control trial (LUMINA-1: NCT03188666; NCT04577820). After reporting encouraging results of the phase II trial in the beginning of 2020, the phase II LUMINA-1 trial was paused due to reports of fatal serious adverse events in the trial during the open-label extension [175]. Results from a phase I trial testing an orally available ALK2 specific inhibitor KER-047 (Keros Therapeutics) were recently reported [176]. KER-047 induced a robust iron mobilization, suggesting a potential application in anaemia. Alternative ALK2 kinase inhibitors are still under early clinical investigation (e.g., BCX9250, BioCryst Pharmaceuticals; BLU-782, Ipsen; Saracatinib, see above), as well as an ALK2-specific neutralizing antibody (Daiichi Sankyo/Saitama, US patent application 10428148). Whether these or other pharmacological strategies for FOP may successfully prevent HO, it remains a risk that toxic side effects will appear. To avoid those, the particular mechanisms mediating the aberrant function of the mutant ALK2 receptor and its functional interaction with activin must be well dissected and targeted specifically, thereby avoiding potential effects compromising the physiological functions of ALK2 (and related TGF-β/BMP receptors) and activins.

One possibility might be to reduce the exposure to these newly developed molecules, by for example, lowering the dose and/or time of administration, while combining drugs to achieve a synergistic effect. Interestingly, molecules (some of them obtained by drug repurposing) have already been developed targeting different steps within the HO pathway (see Figure 2), thereby favouring this approach. Furthermore, one should not forget that safe therapies for FOP may be repurposed to other disease indications as well, thereby increasing their market value. For example, nonhereditary forms of HO or disorders of exacerbated calcification may benefit from FOP-targeting agents.

As the case in FOP, toxicity issues have emerged for several drugs with promising results in preclinical studies and phase I-II clinical stages. Since FOP causes a deep musculoskeletal alteration, the primary aim of drug development has traditionally focused on normalize bone overgrowth in suffering patients. However, FOP involves a number of secondary symptoms, such as brain alterations and cardiovascular complications, which may underlie the severe adverse events observed in late clinical study phases, when individuals are exposed chronically to a drug under investigation. We suggest that the evaluation of drug toxicity in FOP needs to be examined from early preclinical phase studies, using relevant models with patient-derived cells or transgenic animal models. With this regard, the establishment of FOP iPSCs and development of protocols of cell-type differentiation, in combination with advanced in vitro culture systems resembling 3D organ architecture and the contribution of multiple cell types, represent promising tools to identify novel safe and effective FOP drug candidates.

In summary, FOP exemplifies how advances in basic biomedical research pave the road for novel therapeutic approaches. The field is rapidly evolving and new targetable pathways in HO are being discovered. Existing (pre)clinically advanced and approved drugs represent a useful repository of compounds to test in relevant models of FOP. Furthermore, repurposed drugs represent a way to speed up and lower the cost of clinical studies, which are key limiting steps in the market of ultrarare diseases.

## Figures and Tables

**Figure 1 biomedicines-09-00213-f001:**
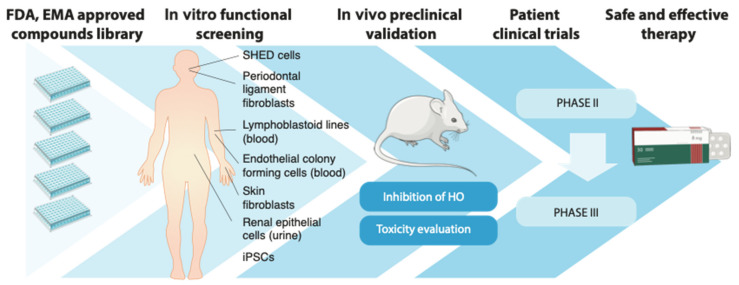
Drug repurposing route in fibrodysplasia ossificans progressiva (FOP). Although other paths are possible (i.e., serendipitous findings) that enable drug repurposing, one common approach starts with commercially available compound libraries containing drugs already approved for their use in humans by the corresponding medical authorities (i.e., US Food and drug administration [FDA], European medicines agency [EMA]). Such libraries are screened in in vitro cell-based assays of FOP, where the mutant receptor ALK2^R206H^ drives a clear functional phenotype (e.g., SMAD1/5 activation or osteoblast/chondrocyte differentiation). Patient-derived cells are very useful in this sense. Selected candidates are subsequently validated in animal models of FOP for their toxicity and effectiveness preventing heterotopic ossification (HO). Because the pharmacological properties of these drugs have been already characterized, lead candidates can immediately enter Phase II and Phase III clinical studies, where disease progression is monitored (for example, by NaF PET/CT scan [82]) before their final approval and authorized application in the clinical practice.

**Figure 2 biomedicines-09-00213-f002:**
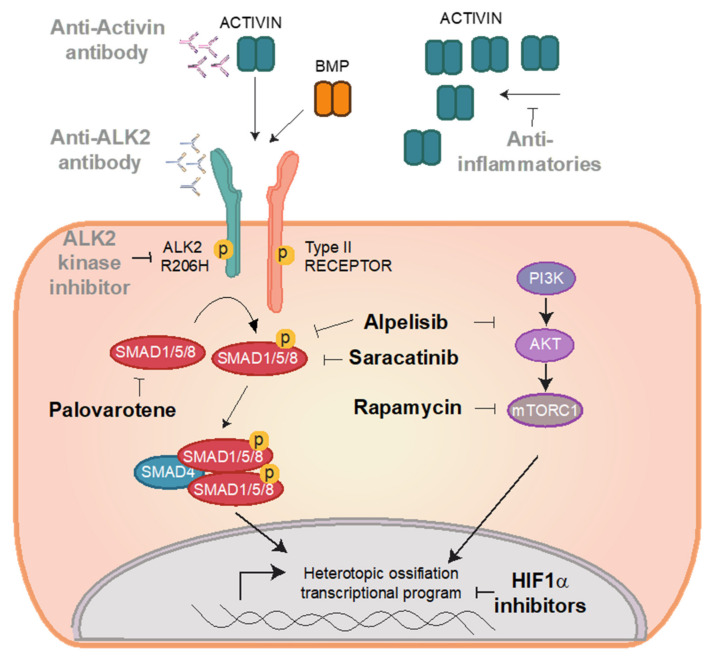
Repositioned strategies for FOP. The mutant ALK^R206H^ bone morphogenetic protein (BMP) type I receptor has been associated with FOP. Compared to the wild-type allele, ALK^R206H^ exhibits leaky responses to BMP ligands as well as neofunctional responses to activin. Binding of BMPs or activins leads to overactivation and phosphorylation of SMAD1/5/8 canonical signalling and non-canonical pathways, which all contribute to HO in bone progenitor cells. Based on this knowledge, a number of targeted therapeutical approaches (depicted in grey) are currently under investigation. Broad-spectrum anti-inflammatories are prescribed to limit the expression of activin in HO lesions. Anti-Activin antibodies (or ligand traps) and anti-ALK2 antibodies are being evaluated to prevent ligand-receptor interaction and/or activation of ALK^R206H^. ALK2 kinase inhibitors aim to block the intracellular kinase activity of ALK^R206H^. In the past recent years, alternative drugs have emerged from drug-repositioning strategies (depicted in bold). Saracatinib inhibits the kinase activity of ALK^R206H^. Rapamycin inhibits mTOR signalling, while alpelisib may inhibit mTOR and SMAD1/5/8 activation simultaneously. Palovarotene inhibits the expression of SMAD1/5/8. HIF1α inhibitors partially block osteo/chondrogenic differentiation transcriptional programs.

**Table 1 biomedicines-09-00213-t001:** Drug repurposing in rare diseases.

Drug Repurposing in Rare Diseases	
Benefits	Disadvantages
Lower investment required	Patent already filed, might require negotiation
Possibility of synergistic effect by targeting different pathways simultaneously	Drug candidates may not be clean and have additional unwanted targets
Compound libraries commercially available	
Drug-like properties already characterized	
Lower number of patients in trials	
Shorter development time	
Limited risk of failure	

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
