# Peer review of "Challenges and Opportunities for Drug Repositioning in Fibrodysplasia Ossificans Progressiva"

_biomedicines, 2021, doi:10.3390/biomedicines9020213_

Round 1
Reviewer 1 Report
The topic is interesting. However, the paper is disorganized and should be reorganized.
Introduction should be shortened. I would limit information to biology and clinical presentation.
Paragraphs "TGF signaling" and "ACVR1/ALK2 signaling in FOP." are disorganized and not focused on FOP. They should be shortened and focused on the topic. Images might help to follow.
"In vitro screening platforms for FOP": this paragraph is partially disconnected with the rest of the paper.
I direct line between signaling pathway through the target drug would be more clear.
Author Response
We appreciate very much the assessment of our manuscript by these two reviewers.
Please, find below a point-by-point answer to the questions raised:
Reviewer #1:
The topic is interesting. However, the paper is disorganized and should be reorganized.
Response: We acknowledge Reviewer #1 for his/her useful comments and suggestions. In order to satisfy the concerns raised, we have adjusted the text and one Figure in the manuscript. We have particularly focused on improving the text flow and connections between sections. Some sections have been shortened and linking paragraphs have been incorporated. Finally, the manuscript has been proofread by English native speakers.
Introduction should be shortened. I would limit information to biology and clinical presentation.
Response: In this revised version we have considerably shortened the introduction according to Reviewer #1 suggestions.
Paragraphs "TGF signaling" and "ACVR1/ALK2 signaling in FOP." are disorganized and not focused on FOP. They should be shortened and focused on the topic. Images might help to follow.
Response: Rather than up-regulated BMP signalling, FOP results from disbalances in TGF-b signaling, both BMP and TGF-b branches. Feedback mechanisms between these two pathways in the context of FOP are only being unveiled in the last years. Furthermore, repositioned drugs may act by targeting these mechanisms. Therefore, we consider that a comprehensive understanding of TGF-b signalling (rather than only BMP or ALK2 signaling) is needed. We cite excellent review manuscripts illustrating these pathways.
In order to improve the text flow, we have rewritten the starting and ending paragraphs in these sections to favour the connection with prior and consecutive sections. In addition, we have rearranged the text to emphasize the role of ALK2 modulating signaling induced by members of the TGF-b family other than BMPs.
"In vitro screening platforms for FOP": this paragraph is partially disconnected with the rest of the paper. I direct line between signaling pathway through the target drug would be more clear.
Response: We have renamed the section and further developed an introductory paragraph, to highlight the need and involvement of in vitro models for FOP in drug development. Moreover, we now refer to in vitro cell based FOP models throughout the text, to improve the embedding of this section. Finally, we have repositioned several elements within Figure 2, to clarify the target of saracatinib and alpelisib.
Reviewer #2:
No comments.
Response: We thank this reviewer for his/her time and consideration of our manuscript as publishable by Biomedicines.
Reviewer 2 Report
In this review, fibrodysplasia ossificans progressiva (FOP) has been discussed in terms of its causative mutation and downstream signaling, in vitro models for drug screening, and possible candidates for effective drugs. Although all these are complicated issues, this review describes them very clearly. It would be helpful for people who are interested in its fundamental mechanisms as well as the most advanced therapeutic challenges in this field.
Round 2
Reviewer 1 Report
The Authors made greta efforts in ameliorating their paper.
It is now more clear to follow